# Accurate prediction of protein assembly structure by combining AlphaFold and symmetrical docking

Mads Jeppesen[1] & Ingemar André [1]✉

AlphaFold can predict the structures of monomeric and multimeric proteins with high accuracy but has a limit on the number of chains and residues it can fold. Here we show that a combination of AlphaFold and all-atom symmetric docking simulations enables highly accurate prediction of the structure of complex symmetrical assemblies. We present a method to predict the structure of complexes with cubic – tetrahedral, octahedral and icosahedral – symmetry from sequence. Focusing on proteins where AlphaFold can make confident predictions on the subunit structure, 27 cubic systems were assembled with a median TM-score of 0.99 and a DockQ score of 0.72. 21 had TM-scores of above 0.9 and were categorized as acceptable- to high-quality according to DockQ. The resulting models are energetically optimized and can be used for detailed studies of intermolecular interactions in higher-order symmetrical assemblies. The results demonstrate how explicit treatment of structural symmetry can significantly expand the size and complexity of AlphaFold predictions.

The functional complexity of cellular processes often requires the association and cooperation of multiple protein subunits in protein complexes. Protein assemblies carry out many of the cell's most fundamental and sophisticated functions, from DNA replication to energy synthesis and molecular motion. As the molecular structure is key to understanding the function of multimeric complexes there is currently significant interest in experimental determination, driven by improvements in cryo-electron microscopy[1], but also in the development of computational methods to predict structure from sequence[2].

Over the last couple of years, we have seen a revolution in our abilities to predict protein structures from deep learning methods. AlphaFold (AF)[3] and AlphaFold-Multimer (AFM)[4] have shown exceptional levels of accuracy in predicting structures of monomers and protein complexes and currently serve as the basis for all of the top-performing models in the latest round of the structure prediction contest CASP[5]. However, AFM is currently limited to smaller complexes as prediction accuracy decreases and memory consumption increases when more chains are modeled[6]. Fundamentally, the prediction of large multimeric assemblies presents significant additional challenges compared to that of monomeric proteins, as it requires the

simultaneous prediction of the organization and interactions of multiple chains in space. The underlying training data to AF/AFM provide rich sources of information regarding the internal structure and interfaces between subunits, but less information regarding the overall organization of chains.

An attractive strategy for building larger complexes is to assemble them from predicted AF/AFM subcomponents. Recently, it was shown that large complexes with 10–30 chains could sometimes be predicted by sequentially assembling AFM-predicted dimeric and trimeric subcomponents by superposition[6]. However, this approach has two major challenges. First, small errors in the prediction of individual interfaces will propagate through the complexes and can result in severe clashes between subunits in the full assembly model. Second, for complexes with more than one unique interface a sequential assembly approach relies on the accurate prediction of multiple interfaces to the same subunit, which can be challenging to extract from AFM.

Moving forward it would be advantageous to develop a method that can iteratively refine all interfaces of the assembly to minimize error propagation and search for interactions not predicted by AFM using molecular docking. This has been demonstrated for

[1]Department of Biochemistry and Structural Biology, Lund University, Lund, Sweden. ✉e-mail: ingemar.andre@biochemistry.lu.se

heterodimeric complexes using two AF-predicted monomers and rigid-body docking methods[7,8]. However, for large assemblies, this approach has not been explored. This is likely due to the very large number of degrees of freedom involved in the search for optimal subunit placements, leading to a computationally intractable optimization problem. Nonetheless, for multimeric assemblies displaying structural symmetry the degrees of freedom can be substantially reduced[9,10], making the combined AF/AFM docking approach tractable.

Many large protein complexes are either fully symmetrical, display local symmetry, or are quasi/pseudo-symmetrical (Fig. S1). Symmetry also becomes more prevalent as the protein complexes grow larger (Fig. S2). The evolution of symmetry in large protein complexes has facilitated the emergence of many shapes such as rulers, rings, and containers that are uniquely important for many functions. It is also critical for allostery, cooperativity, and multivalent binding[11–14].

Here we present a strategy to predict the structure of large symmetrical complexes from AF or AFM subcomponents with high

accuracy by combining it with an all-atom symmetrical docking method (Fig. 1a). We recently presented an efficient atomistic docking algorithm for heterodimeric docking called EvoDOCK[15]. EvoDOCK uses a differential evolution algorithm for efficient sampling of rigid-body space coupled to a Monte Carlo approach for local search optimization of all-atom interactions. In this study, we extend it to symmetry and use it to build and refine complex homomeric symmetrical assemblies built from subcomponents predicted by AFM. We demonstrate our method on a benchmark containing large protein assemblies from the most complex symmetrical systems in nature, the cubic symmetry group. The cubic symmetry group consists of the tetrahedra (T), octahedra (O), and icosahedra (I) with 12, 24, and 60 chains respectively in the simplest homomeric case (Fig. 1b). These complexes form spherical structures and are often involved in a variety of biological functions including the storage of genomes for many viruses. We show that our method can predict the structure of cubic symmetries from sequence at an atomic level, providing energetically optimized models of very large assemblies.

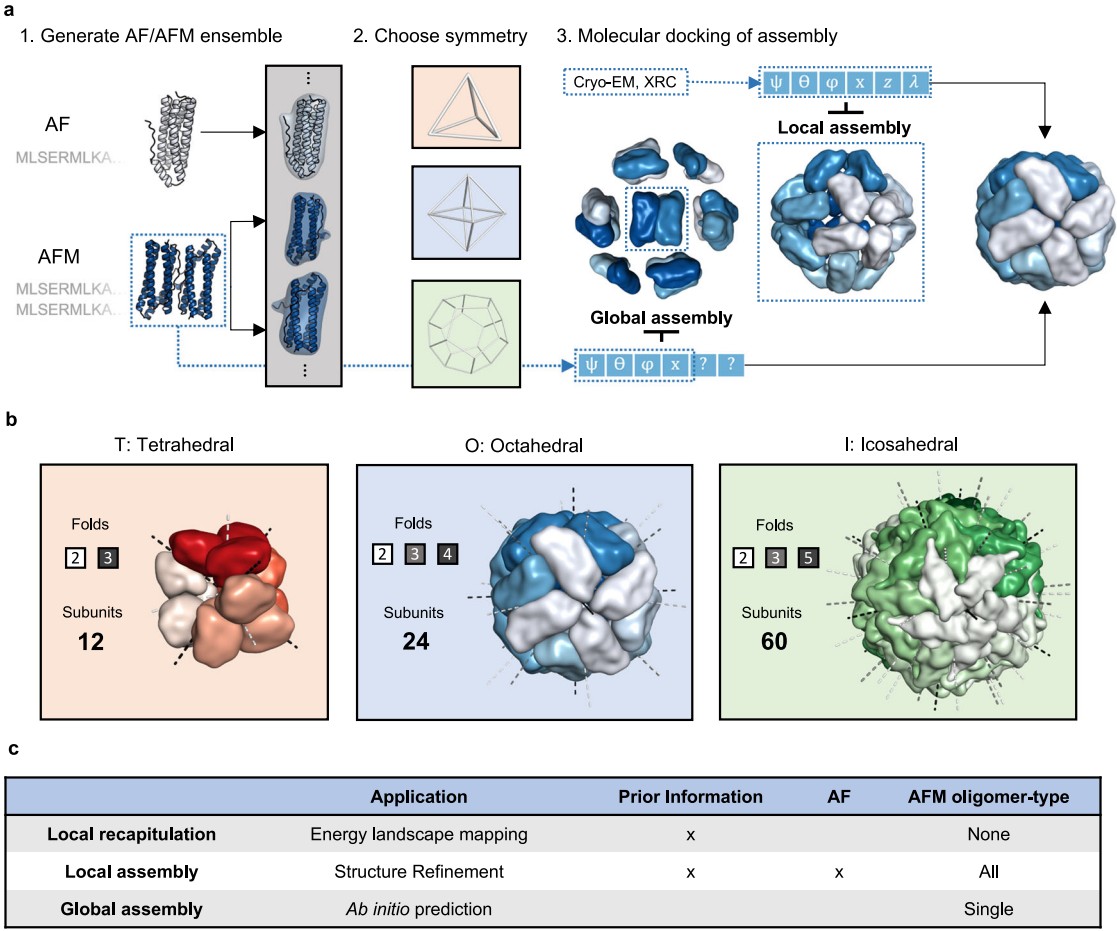

Fig. 1 | Schematic representation of the assembly prediction method and the cubic symmetry group. a Overview of the assembly prediction method. Step 1: AF (top-left) and/or AFM (bottom-left) are run on sequences of the target protein complex to produce an ensemble of input subunits. Step 2: A cubic symmetric model of either a tetrahedral (T), octahedral (O), or icosahedral (I) type is chosen to model the target protein complex. Step 3: The subunits are placed into the chosen symmetry and molecular docking is used to search for native rigid-body parameters describing the cubic protein complex ($[\psi, \Theta, \varphi, z, x, \lambda]$, see also Fig. 2b). Two approaches can be used to aid the modelling of the structure shown by dashed lines in the figure. With access to some structural knowledge to set the parameters, which can be derived from a variety of sources such as cryo-EM or X-ray crystallography (XRC), the structure can be reassembled (local assembly) by searching for

local optimal values of the parameters ($\psi, \Theta, \varphi, x, z, \lambda$). In the absence of any structural information, the structure can be completely assembled (global assembly) from sequence using a combination of local docking on parameters derived from AFM parameters ($\psi, \Theta, \varphi, x$) and global docking on the rest ($z, \lambda$). b The three cubic symmetries that are modelled. Tetrahedral (T) structure having 2- and 3-fold symmetry and containing at least 12 subunits. Octahedral (O) structure having 2-, 3- and 4-fold symmetry containing at least 24 subunits. Icosahedral (I) structure having 2-, 3- and 5-fold symmetry containing at least 60 subunits. c Table of the three assembly approaches, their applications and differences in modelling. Local recapitulation is a variant of local assembly where the native structure is used directly instead of AF/AFM predictions.

## Results

The general methodology for predicting large protein complexes from AF/AFM subcomponents is presented schematically in Fig. 1a. Sequences of the target protein complex are used as inputs to AF/AFM which in turn are used to generate an ensemble of different candidate subunits for the target complex (Fig. 1a, step 1). One of three cubic symmetries is chosen to model the target structure (Fig. 1a, step 2) and a symmetric version of EvoDOCK is then used to dock the assembly to produce a final energetically optimized model by optimizing the rigid-body parameters describing a cubic system (Fig. 1a, step 3).

Here we distinguish between three approaches to model the target structure which we call *local recapitulation, local assembly* and *global assembly* and their applications and differences are highlighted in Fig. 1c. In the *local recapitulation* approach, the degrees of freedom associated with rigid body and sidechains are optimized starting from a native assembly model and bound capsid subunit. This type of simulation can be used to characterize the energy landscape of assembly and improve energetics of a structural model. In the local assembly simulation, the initial rigid body orientation is also taken from a starting model, but the subunit structure is predicted with an ensemble from AF and AFM. This type of simulation can be used to build a model based on a template from Cryo-EM, X-ray crystallography or other computational methods. In the absence of structural information, a larger parametric space must be sampled as no templates are available. In the global assembly approach, the full complex is predicted ab initio, that is, directly from sequence using only single type (dimeric, trimeric etc.) AFM oligomer predictions as a basis. Through AFM, some of the rigid body parameters can be estimated and locally optimized (Fig. 1a, blue dashed box) while others must be globally optimized.

A benchmark of protein structures with cubic symmetry was constructed to evaluate the method, consisting of assemblies with tetrahedral, octahedral, and icosahedral symmetries. Requirements on experimental resolution, structural diversity, and monomer size were used to select the final set. In addition, we limit ourselves to systems where AF/AFM can accurately predict monomer and subcomponent structures (see "Methods" for the full selection procedure).

The final benchmark contains 27 cubic systems (Table S1), 9 from each of the three symmetry groups. AFM has not been trained on multimers of cubic symmetry, which all have more than 9 chains. However, subunits may be homologous to smaller oligomers. The sequence identity to smaller oligomers that may have been used to train AFM (the training set data has not been released) is presented in Table S1. 12 of the systems have low homology (30% sequence identity) to any smaller oligomer found in PDB released before the cutoff for the training of AFM. This categorization allows us to study the influence of homology on the accuracy of AFM predictions.

Accurate prediction of assembly structure required an extension of the EvoDOCK approach to cubic symmetry and optimization of conformational sampling for complex and large symmetrical systems, described in the next section. This is followed by a section on how to utilize AF and AFM to generate structural ensembles required for the assembly simulation. We test our methodology on increasingly more difficult scenarios. In the first scenario, we show that the native assembly structure can be recovered in local recapitulation experiments using the backbone of the native monomer. In the second scenario, we show that the assembly structure can be recovered in local assembly experiments using subcomponent structures predicted by AF/AFM. In the third scenario, we demonstrate that the structure of cubic symmetrical systems can be predicted directly from sequence without prior information on the rigid body orientation in global assembly experiments. Finally, as the benchmark is limited to systems that produce accurate and symmetrical AF/AFM predictions, a broader spectrum of cubic subcomponents with AF/AFM is predicted and we analyze what fraction we can expect to be successfully predicted by the method.

### Assembly of proteins with cubic symmetry using symmetrical docking

The symmetric docking step in this study is built on an extension of the EvoDOCK method[15] to symmetry. EvoDOCK is based on a memetic algorithm that combines a differential evolution method coupled to a Monte Carlo local docking search and is presented schematically in Fig. 2a.

A population of individuals, each containing a randomly chosen backbone from an ensemble and six randomly chosen rigid body parameters, are initialized. The backbones and rigid body parameters of the individuals are optimized through a series of generations containing four steps: *Evolution, Sliding, Local search,* and *Selection*. During Evolution, the individuals share optimal parameters through mutation and recombination events. This drives the individuals towards more optimal solutions in the population but can introduce suboptimal energies. In the Sliding and Local search step, energies are refined by first sliding the subunits towards each other followed by an optimization of the rigid body parameters using a Monte Carlo and minimization search strategy. In the Selection stage, each optimized individual is compared to its predecessor and is either continued or reverted by selecting the one with the best energy. Occasionally a new backbone is inserted into the individual from the ensemble (BB trial) and the Sliding, Local search, and Selection stage is repeated. After several generations, the whole structure is energy refined where the backbone and sidechains are simultaneously optimized and a final model is output. The optimization is guided by the all-atom energy function of Rosetta[16,17] and uses the symmetry machinery of Rosetta to model the structure[10].

To adapt EvoDOCK for cubic symmetry we changed the six rigid body parameters based on a heterodimeric docking system to six based on cubic systems (Fig. 2b). Conformational sampling in cubic symmetry is complex due to the high packing density of subunits, the high degree of shape complementarity between chains, the presence of multiple protein–protein interfaces in the assembly and the fact that small changes in parameters can lead to drastic changes in overall assembly structure. To address these issues several improvements to the EvoDOCK methodology were made. For computational efficiency, the complete assembly structure is not modeled but rather a subsystem of chains that contains all the interactions required to calculate the energetics of the complete system (Fig. 2c). To maintain the integrity of this subsystem and enable the calculation of the whole structure energy, rigid body parameters are not sampled freely but constrained within an interval (Fig. 2d, "Methods"). This also has the benefit of making conformational sampling more efficient. During the simulation, one of the types of interfaces (2, 3, 4- or 5-fold) present in the cubic assembly may be correctly identified, while others are suboptimal. We implemented a sampling approach where the optimal interfaces can be kept, while primarily improving contacts to the other interfaces using optimized sliding moves (Fig. 2e, "Methods").

A fundamental challenge with sampling in cubic symmetry is to identify clash-free and energetically realistic subunit packings. This is particularly difficult using an all-atom energy representation, which produces very complex energy landscapes where tiny changes in orientation can lead to large atomic clashes. To address this problem, we designed a subroutine during the Local search to identify clash-free packing orientations that can be further improved in the all-atom optimization step. The subroutine does multiple rigid body perturbations guided by a score function which we call CloudContactScore (CCS) (Fig. 2f, "Methods"). Internally in CCS, the assembly is represented as a cloud of points consisting only of the surface atoms of the backbone of the subunits. This fast-to-calculate representation is used

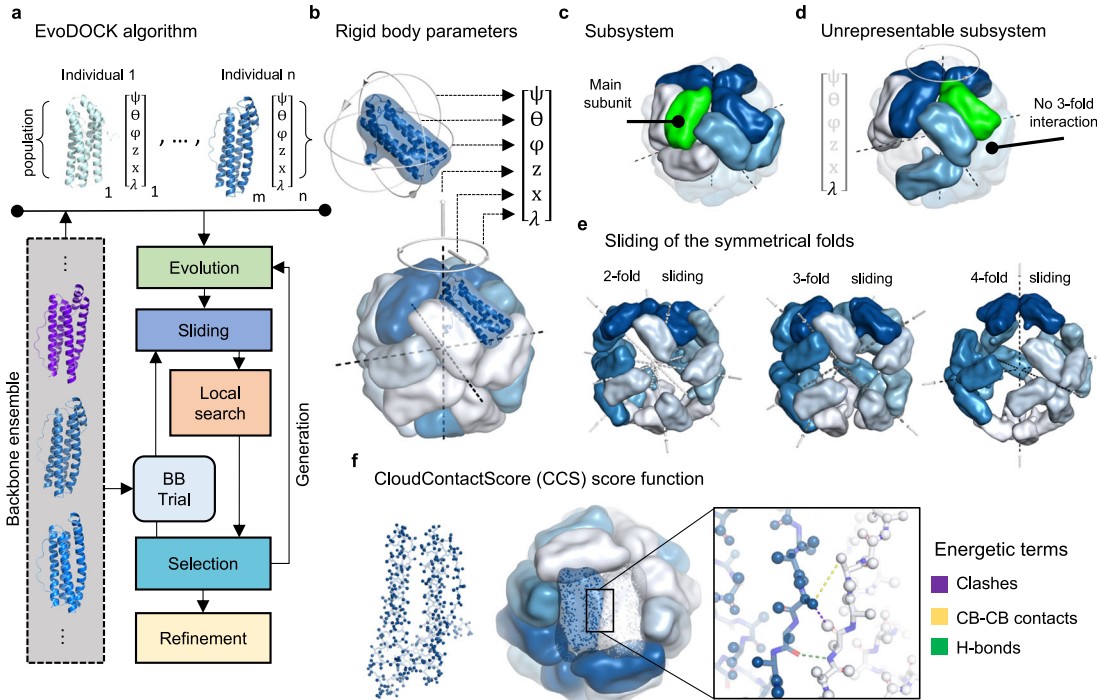

**Fig. 2 | Symmetric EvoDOCK. a** The EvoDOCK algorithm as described in the main text. **b** The cubic rigid body representation is composed of three parameters that control the rotation around each subunits center of mass ($\psi$, $\Theta$, $\varphi$), and three that control the orientation of the entire assembly (z, x, $\lambda$). **c** The subsystem (bright surfaces) is used internally to represent the entire assembly. The subunit in green is the main subunit from which all energetic interactions are calculated. **d** The integrity of the subsystem can be lost due to large perturbations of z, x, and $\lambda$. Here $\lambda$ is perturbed beyond its bounds so that the whole structure energy will be wrong

as the 3-fold interaction is not seen by the main subunit. **e** Sliding axis of the 2-, 3- and 4-fold axis during sliding in an octahedral case. When sliding along the 2-fold for instance, the 2-fold interface is kept fixed while the 3- and 4-fold interface contacts are improved. **f** Left: Atoms on the surface (shown in dark blue) of each subunit's backbone and first atom (CB) of the sidechain (shown in light blue) as represented internally in CloudContactScore (CCS). Middle/Right: Terms for the score function used in CCS to evaluate interchain interactions.

to quickly guide structures towards fewer clashes, more interface hydrogen bonds, and better backbone contacts.

To test the proficiency of the methodology a local recapitulation experiment was carried out for all the systems in the benchmark using the bound experimental structure as a starting point. The 6 parameters describing the rigid body orientations of the subunits in the assembly were randomly sampled uniformly in a broad range centered around the values found in the native system with an average RMSD of 11 Å. 100 independent EvoDOCK simulations with a population size of 100 each were started from these configurations and run for 50 iterations. In 22 out of 27 systems the lowest energy model after recapitulation had an RMSD over the subsystem below 2.0 Å, while the remaining had 3.6 Å to 9.4 Å (Fig. S4). Note that the RMSD values cannot be 0.0 Å in all cases, because some of the experimental structures are not fully symmetrical. These results demonstrate that most cubic symmetrical systems can successfully be recapitulated if the native subunit structure is known. Differences compared to the native structure were primarily due to the identification of structures with alternative minima in the energy landscape, rather than inefficient rigid body sampling. Fig. S3 collects some examples from the different sampling scenarios (Fig. 1c) where issues with the energy function, conformational sampling, and energy refinement causes predictions to fail.

**Prediction of subcomponent structures from sequence using AF/AFM**

Prediction of the structure of cubic systems from sequence requires prediction of subcomponent structures with AF and AFM. Predicted subcomponents are then used as inputs for the symmetric docking simulations. Using predicted subcomponents is significantly more challenging for two reasons: First, the predicted backbones will not be

perfect, requiring different candidate backbones to be sampled and optimized. Second, there are parts of the structure that can only be predicted correctly in the presence of the full assembly. The N- and C-termini are often involved in the interfaces of cubic assemblies. However, AF/AFM generally cannot predict termini well and the presence of misfolded segments can prevent the formation of the correct assembly in the docking simulation. Nonetheless, a balance between removing residues with a high degree of prediction uncertainty and keeping residues that are important for interface formation must be struck. The same strategy can be employed to remove internal loops with uncertain conformations that can hamper assembly, but this was not necessary for the current benchmark.

The strategy used to produce alternative subcomponent backbones as inputs to EvoDOCK is shown schematically in Fig. 3a–c. The goal was to produce backbone ensembles with some diversity while keeping them close to the predictions by AF/AFM. We initially experimented using Rosetta[18] to resample backbones from AF/AFM predictions as previously done in EvoDOCK[15] and elsewhere[19,20] but found that backbones sampled in this manner have significantly higher RMSD to the native backbones compared to the raw output from AF/AFM. The AF/AFM predictions were therefore used directly, employing random seeds to produce different conformations. One of the strengths of AF/AFM is the ability to evaluate its prediction confidence through predicted values of the local structure quality metric LDDT[21] (pLDDT) and the whole structure quality metric TM-score[22] (pTM and ipTM for interfaces). These metrics were used to identify a diverse set of near-native initial backbone ensembles (Fig. 3a, "Methods").

We utilize this initial backbone ensemble set to decide what terminal residues to remove for the docking simulations. Three metrics were used (Fig. 3b, "Methods"): The average residue pLDDT,

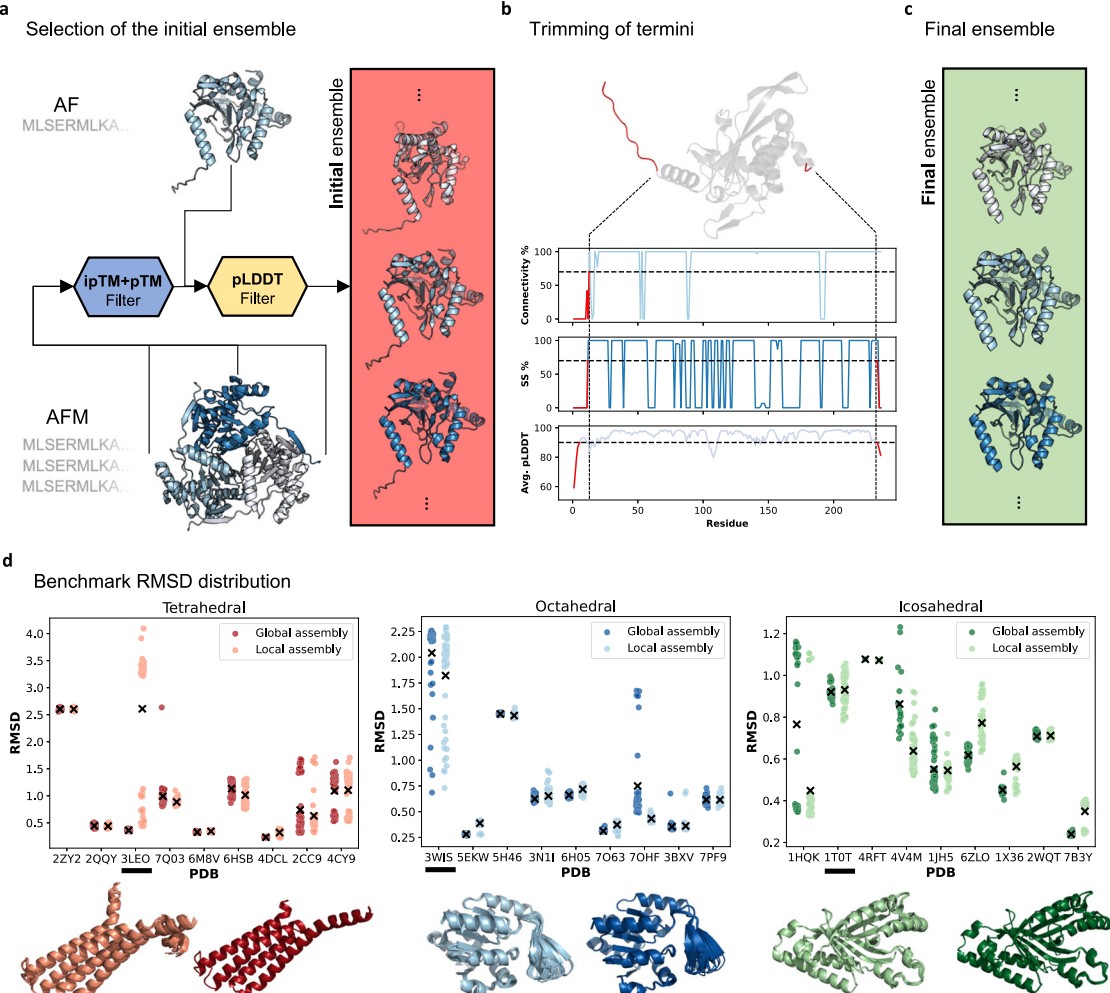

**Fig. 3 | Ensemble generation strategy. a** AF predictions are filtered based on their pLDDT scores and AFM predictions are filtered based both on their pLDDT and ipTM+pTM scores (AFMs model confidence score: $0.8 \cdot$ ipTM + $0.2 \cdot$ pTM) to generate an initial ensemble (red box). **b** To determine how many residues to trim at the N- and C-termini, the initial ensemble is collectively used to produce average values of three metrics per residue: connectivity (connectivity %), secondary structure propensity (SS %) and average residue pLDDT (Avg. pLDDT). Threshold values (dotted horizontal lines: 90 for Avg. pLDDT, 70% for connectivity %, and 70% for SS %) for the metrics are set and terminal residues were removed from the ends until all of the metrics goes beyond their respective threshold values (indicated by the red lines). **c** The final ensemble, which is created by removing residues as described in b, contains subunits for the target structure of equal sequence length (green box). **d** Ensemble RMSD distribution to the monomeric native structure for all benchmark structures for both local assembly (lighter color) and global assembly (darker color) (n for global/local assembly: 2ZY2: 53/61. 2QQY: 71/79. 3LEO: 121/62. 7Q03: 38/55. 6M8V: 54/57. 6HSB: 42/58. 4DCL: 29/75. 2CC9: 45/57. 4CY9: 54/57. 3WIS: 52/50. 5EKW: 120/60. 5H46: 105/53. 3N1I: 50/56. 6H05: 80/62. 7O63: 88/56. 7OHF: 46/73. 3BXV: 33/50. 7PF9: 37/52. 1HQK: 30/57. 1T0T: 44/54. 4RFT: 1/5. 4V4M: 18/54. 1JH5: 48/59. 6ZLO: 51/50. 1X36: 22/56. 2WQT: 30/58. 7B3Y: 53/61). The "x" symbol shows the mean. Below each subfigure, two examples (underlined) of the structural ensembles superimposed for the local (left) and global (right) assembly are shown. Source data are provided in the Source Data file.

the secondary structure propensity, and the residue connectivity to the rest of the structure. The pLDDT score evaluates the AF/AFM's confidence on a per-residue basis, while the secondary structure propensity and connectivity provide measures of the expected degree of flexibility and residue-residue interaction density. Terminal residues are removed from the ends until all of the metrics goes beyond their respective threshold values (dashed lines in Fig. 3b, "Methods"). The final ensemble, which is used as inputs to the symmetric docking simulations contains subunits for the target structure of equal sequence length (Fig. 3c). The resulting RMSD distribution compared to the native subunit are shown in Fig. 3d for all benchmark cases.

### Local assembly of proteins with cubic symmetry from predicted subunit structures

To test how well the methodology works on subcomponent structures predicted by AF/AFM we carried out the types of rigid body perturbation local assembly described previously, in which the rigid body

parameters are uniformly sampled around values from a template symmetry. In our case, the parameters were calculated with the native structure as a template. However, initial rigid body parameters can also be taken from the structures of a homologous protein or estimated from a lower resolution Cryo-electron microscopy/X-ray crystallography structure in the context of model refinement.

Ensembles for subunit structures were generated as previously described using a combination of monomeric AF and AFM predictions with different oligomeric states (see "Methods"). The ensembles were used as inputs to 100 independent EvoDOCK simulations with a population size of 100. 100 of the best models according to the interface score (Iscore) from the EvoDOCK simulations were selected and refined using a symmetric energy refinement method in Rosetta[23] (See "Methods"). 5 final predicted models are output by the method by clustering all 100 refined models based on the predicted rigid body parameters into 5 sets and then selecting the best model in each set according to the Iscore. The 5 models are ranked according to the

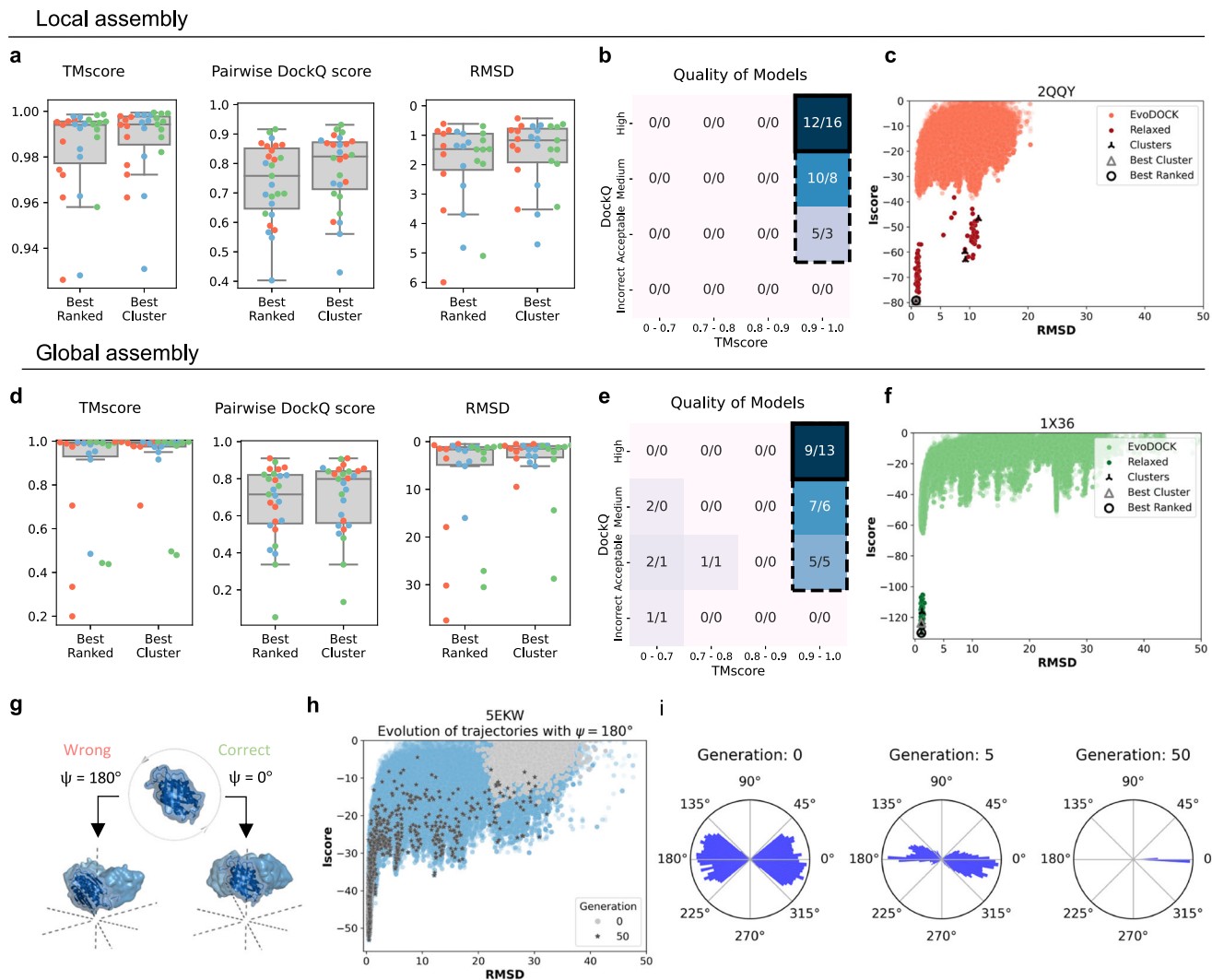

**Fig. 4 | Results of the local and global assembly experiments. a–c** Results of the local assembly experiments. **a** TM-score (*n* = 27), Pairwise DockQ score (*n* = 27), and RMSD (*n* = 27) for all benchmark structures (T = red, O = blue, I = green). Data are represented as boxplots (grey) with the median at the center, 25th percentile at the lower bound, 75th percentile at the upper bound, and whiskers indicating the minimum and maximum values. **b** Classification of the results using TM-score and Pairwise DockQ score. The outer dashed line delineates successfully predicted structures and the inner black box the highly accurate structures. **c** Example of an energy landscape (RMSD vs Iscore). **d–i** results of the global assembly experiments. **d** TM-score (*n* = 27), Pairwise DockQ score (*n* = 27), and RMSD (*n* = 27) for all benchmark structures (T = red, O = blue, I = green). Data are represented as box-plots (grey) with the median at the center, 25th percentile at the lower bound, 75th percentile at the upper bound, and whiskers indicating the minimum and

maximum values. **e** Classification of the results using TM-score and Pairwise DockQ score. The outer dashed line delineates successfully predicted structures and the inner black box the highly accurate structures. **f** Example of an energy landscape (RMSD vs Iscore). **g** Two possible orientations of the oligomer structure in cubic symmetry related by a 180° turn through the $\psi$ parameter. **h** Orientation optimization ($\psi$) of the subunits within the assembly of the model 5EKW. Models that are started in the wrong orientation are shown at the beginning (generation 0) and at the end (generation 50). Most of the models started in the wrong have learned the correct orientation at the end of the simulation. **i:** Same simulation as in H but showing snapshots of the distribution of orientation angles ($\psi$) as function of the generations. Source data for **c, f, h, i** are provided in the Source Data file, and the rest in Supplementary Tables 3 and 4.

Iscore, and here we analyze them with respect to the best-ranked model (Best Ranked) and best model among all the clusters (Best Cluster) according to three metrics: TM-score[22], DockQ[24], and RMSD.

For all 5 models, we calculated the TM-score, average pairwise DockQ score across the symmetrical interfaces (Pairwise DockQ score) and the RMSD (Fig. 4a and "Methods"). We find median values for the Best Ranked/Cluster model as TM-score: 0.99/0.99, Pairwise DockQ score: 0.76/0.82, and RMSD: 1.5/1.2 Å. We here define successful predictions as predictions having at least acceptable quality in their Pairwise DockQ score and a TM-score of at least 0.9 and highly accurate predictions as having high quality in their Pairwise DockQ score and a TM-score of at least 0.9 (Fig. 4b). Under that definition, 100/100% of models are successfully predicted and 44/59% have highly accurate

predictions. An example of an energy landscape for 2QQY is shown Fig. 4c and all energy landscapes and metric values are shown in Fig. S5 and Table S3.

## Global assembly prediction of the structure of cubic systems from sequence

The previous results demonstrate that with reasonable values for the rigid body parameters describing the symmetry of the cubic system, the method can successfully predict the structure of cubic complexes. In this section, we will attempt to predict the structure without any prior information or assumptions about the rigid body orientations in global assembly experiments. The basis of the prediction is that we can predict a single oligomeric subcomponent from AFM (dimer, trimer,

tetramer, or pentamer) and use it as the starting point for the cubic assembly prediction. Ensembles of subunit structures were generated as previously described with AFM. But in addition, symmetry information was extracted from the predicted oligomer, enabling some of the rigid body parameters in the docking simulation to be estimated (Fig. 2b; $\psi, \Theta, \varphi, x$, see "Methods"). Without these initial estimates the correct assembly are difficult to predict. In Fig. S10 we show results of 3 simulations where the subunits structure is correctly predicted but the oligomers are not, resulting in inaccurate assemblies. In the EvoDOCK simulation, we sample around the individual values found in the AFM oligomer predictions. The remaining rigid body parameters (Fig. 2b; $z$, $\lambda$, see "Methods") fully unknown, parameters were sampled uniformly. To get a good starting model to initiate EvoDOCK, the AFM predictions are docked along their respective symmetrical fold (Fig. 1b, see "Methods"). But given that certain parameters are constrained, there are two ways to place the oligomer, related by a 180-degree flip perpendicular to the symmetry axis (Fig. 4g). It is possible to run two independent simulations starting from each orientation, but we can also let EvoDOCK learn the right orientation during the simulation. Figure 4h, i shows an example where the global assembly simulation is initialized with equal number of correct and wrong orientations. At the end of the simulation the correct orientation has taken over (Fig. 4h, i) on the basis on its superior energy. The result of the global assembly benchmark was generated with this approach.

We clustered the results into 5 final models as described previously and calculated the TM-score, Pairwise DockQ score, and the RMSD (Fig. 4d). We find a median value for Best Ranked/Best Cluster as TM-Score: 0.99/0.99, DockQ score: 0.72/0.80, and RMSD: 1.6/1.5 Å. Using the same definition for successful and highly accurate predictions as previously, it is found that 78/89% of models are successfully predicted and 33/48% have highly accurate predictions (Fig. 4e). An example of the resulting energy landscape of 1X36 is shown in Fig. 4f, and all energy landscapes and metric values are shown in Fig. S6 and Table S4. In Fig. 5a, the predicted structures of the same examples are shown overlaid on their native structures. Taken together, the result demonstrates that accurate prediction of the structure of assemblies with cubic symmetry can be predicted with high accuracy.

Knowledge of the right orientation improves the results slightly. When the simulations are started with the correct orientation it is found that 85/89% of models are successfully predicted (Figs. S7, S8 and Table S5), compared to 78/89% when the orientation is predicted. Figure S9 shows examples of where the wrong orientation takes over in the simulation for two icosahedral virus capsid predictions. In both cases, however, the correct orientation can be established based on the expected electrostatic charge distribution in virus capsids and modelling them exclusively in this orientation significantly improves the results (Fig. S9).

## The general applicability of the method
The methodology presented so far relies on the ability of AF/AFM to produce accurate starting models for symmetric docking simulations. To get insight into the general applicability of the method, we estimated the fraction of cubic systems that can be predicted with sufficient accuracy by running AF and AFM on 111 sequences from cubic systems (see "Methods"). For each sequence, we attempted to predict the structure of the monomer as well as the 2-3 types of unique symmetric oligomers present in each cubic symmetry type ($T = 2/3$, $O = 2/3/4$, $I = 2/3/5$). We define an acceptable solution as a prediction within 2 Å RMSD to the native structure. Figure 5b shows how many structures can be predicted within this threshold for AFM while Fig. 5c shows how many interfaces can be predicted for each cubic assembly by AF.

In 78% of the cases, at least one AF structure can be predicted and in 72% of the cases, at least one AFM interface can be predicted. These results demonstrate that in most cases we can expect AF and/or AFM

to produce acceptable inputs for EvoDOCK. Interestingly, in 50% of the cases where at least one interface can be found, AFM cannot acceptably predict all three unique interfaces of a cubic assembly correctly. This highlights the need to use a search algorithm that can find additional interfaces outside of AFM to predict the structure of complex symmetric assemblies as we present here.

In this study, we have benchmarked only highly accurate AF/AFM predictions (≥90 pLDDT for AF and pLDDT ≥ 90 and ipTM+pTM ≥ 0.9 for AFM. ipTM + pTM is AFMs model confidence score: $0.8 \cdot ipTM + 0.2 \cdot pTM$). Figure 5d shows the distribution of pLDDT and ipTM+pTM from the AFM predictions. We find that 58% of the structures have this required accuracy. Figure 5e shows the distribution of pLDDT from the AF predictions. We find that 82% have this accuracy. For the benchmark set, we required at least one AF and one AFM prediction to be above this threshold. The percentage of structures passing these filters is 57%. It suggests that a large fraction of proteins with cubic symmetry can be predicted, even with this stringent threshold. As structure predictions evolve, this number is expected to increase. Furthermore, not limiting the AF/AFM to templates deposited before the release date of the benchmark structures as done here also suggests this number can be higher in practical cases. By combining the fraction of cubic systems that pass the required threshold (57%), with the fraction of predictions that pass the structure quality threshold after EvoDOCK we estimate that the current approach can successfully predict around 57%, considering both the best-ranked or best cluster model of all homomeric cubic systems in a local assembly experiment. In a global assembly experiment, we estimate that around 44% and 50%, considering either the best-ranked or best cluster model, respectively, of all homomeric cubic systems to be successfully predicted by our method.

A caveat with the analysis presented here is that sequences of experimentally solved cubic proteins may be biased towards having more homology to other proteins in the PDB than average. The fraction of correctly predicted cases with <30% sequence identity to any oligomer in PDB before the cutoff date used for AFM training, is 67% for the best-ranked model, while 78% for the complete benchmark. Also, the fraction of cubic systems that can be predicted above the quality threshold for AFM with our method is lower if the homology is low (<30%), 33% compared to 85% for sequences above 30% identity (Fig. 5f). Within this low homology regime, the expected success rate drops to 22% for the predictions in this category, largely due to the reduced quality of AFM predictions. However, in many real-case scenarios, predictions will be carried out for systems with significant homology to other known structures. In addition, in real-case scenarios, there are no limitations on which structural templates can be used in AF/AFM predictions, as opposed to the limitations in the AF/AFM predictions run in this study (see "Methods").

## Discussion
Complex protein assemblies with higher-order symmetry are currently difficult to predict with deep learning methods. Symmetry puts important constraints on the structure of homomeric assemblies but is currently not directly modeled with an approach like AFM. Nonetheless, symmetry often emerges from multimer predictions of homomers so that smaller complexes can be accurately predicted. Protein complexes with cubic symmetry are far beyond what can be currently predicted with AFM due to the limit on the number of residues and chains in the current implementation[4]. The method presented here enables atomic-resolution prediction of highly complex protein assembly structures with tetrahedral, octahedral, and icosahedral symmetry by combining the capabilities of AF and AFM to model subunits and small oligomers from sequence and the capabilities of a symmetric docking protocol to model complex structural symmetry. A fundamental benefit of this approach is that it produces models that are optimized in terms of intermolecular interactions,

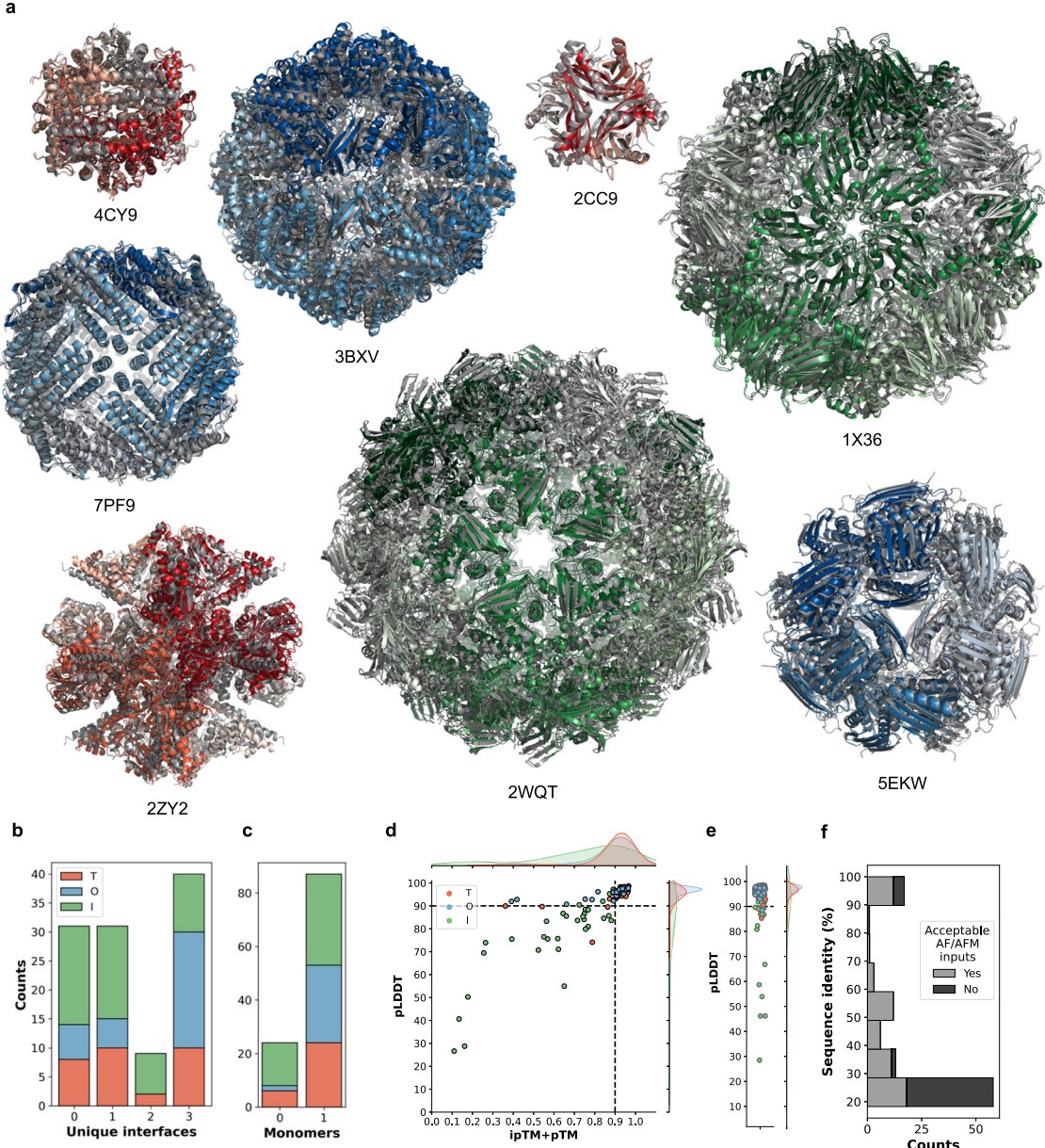

**Fig. 5 | Ability of AF/AFM to predict predictions of sufficient accuracy for EvoDOCK. a** Predicted structures shown in colors (T, red; O, blue; I, green) overlayed on the native structure in grey. **b** The number of unique interfaces AFM can predict per assembly in the set of 111 cubic sequences. **c** The number of monomers AF can predict in the set of 111 cubic sequences. **d** Scatterplot of AFM pLDDT vs ipTM+pTM for the 111 sequences, with kernel-density estimates shown above and to the side of the figure. Dashed lines are drawn at pLDDT = 90 and ipTM+pTM = 0.9. **e** Scatterplot of AF pLDDT values in the set of 111 cubic sequences. **f** Histogram over the count of sequences above and below the quality threshold for AF/AFM as function of sequence identity to oligomers in the PDB before cutoff for training of AFM. Source data for **b**–**f** are provided in the Source Data file.

which makes them a suitable basis for detailed analysis at the atomic level. In addition, energies provide an orthogonal quality metric that can be used to distinguish between alternative models and can be used to resolve unphysical arrangements of chains that can result from assembly approaches based on superimposition of oligomeric subsystems.

In this study, we have limited ourselves to the most complex symmetrical protein structures in nature, the cubic symmetry group. Nonetheless, the method can readily be extended for other types of symmetrical systems including those with cyclic, dihedral, and helical symmetry by optimizing against a different set of rigid body parameters, using the Rosetta symmetry machinery[9]. The approach could handle heteromeric cases as well, such as icosahedral protein capsids, by predicting heteromeric asymmetric units using AFM and using it as

input for symmetric docking, although this must be tested in further benchmarking studies. We also anticipate that the same concept could be utilized to handle quasi-symmetric[11] capsid systems with triangulation numbers higher than 1.

The method described here is limited by the accuracy of AF and AFM. We demonstrate that AF/AFM can accurately model the monomeric and oligomeric subsystems for a high fraction of cubic systems. As AFM is continuously improved[25] and additional variations of AFM are introduced[5], we expect a larger fraction of models to pass the quality threshold for accurate assembly by EvoDOCK. For example, results in CASP15[5] suggest that improvements in multimer predictions can be made by introducing more variation in inference by using dropouts in AFM followed by ranking by the ensembles with predicted quality metrics[26]. Such ensembles can readily be used with EvoDOCK.

Our approach only requires that one of the three main interfaces in a cubic system can be predicted by AFM, and this is typically the case. This is a benefit compared to a sequential assembly approach that necessitates accurate prediction of multiple types of interfaces by AFM.

In virus capsid structures, N- and C-terminal segments are often involved in interchain interactions and may form important parts of protein-protein interfaces within the capsid. Our method does not currently model these segments. Traditional loop-modeling methods constrained by cubic symmetry could be used to complete the assembly structure. If the terminal segments reach over between different oligomeric subcomponents (domain swapping for example[27]), the current approach will fail. In that scenario, the two subcomponents would have to be modeled together in AFM. Using these more complex subsystems would require further method development.

We anticipate that the methodology presented here could be used to study cubic assemblies in several different modeling scenarios. With the local experiments, the energy landscapes of native assemblies could be investigated to understand the relative importance of subunit interfaces to the overall stability of the protein. Such experiments can also be used to model the effect of mutations and to investigate assembly mechanisms of cubic assemblies. Local assembly experiments can also be used to build models of evolutionary-related assemblies, by modeling the subunit structures with AF and docking them with EvoDOCK. Another application of symmetric EvoDOCK is the refinement of structures against experimental data. EvoDOCK is implemented based on pyrosetta[28], which can readily utilize a wide range of experimental constraints[18], including cryo-electron densities[29]. Finally, the methodology can be used to predict structures of cubic assemblies with unknown structures. This will be particularly useful for icosahedral virus capsids. Estimates suggest that there are around $10^{31}$ viruses on the planet[30], and we can hope to experimentally characterize only a fraction of these systems. Nonetheless, many protein capsid proteins are substantially more complex than the homomeric systems studied here, consisting of many different types of subunits[27,31], having quasi/pseudo-symmetry[32] and consisting of symmetry breaking elements[33] and membrane anchoring. Predicting the structures of more complex biological assemblies will require more sophisticated tools than presented here but will likely require explicit treatment of symmetry and simulations of subunit assembly as we describe in this study.

## Methods

### Prediction with AlphaFold2 and AlphaFold-Multimer

For each PDB the release date in the Protein Data Bank[34] was recorded. AlphaFold 2 (2.2.2) was run setting the --max_template_date flag to be the day before the release date of the PDB and the --model_preset to be either *monomer* for AF or *multimer* for AFM. AF and AFM was run as follows:

```
alphafold --fasta_paths=<fasta file path> --model_preset=<monomer/multimer> --output_dir <output directory> --db_preset = full_dbs --use_gpu_relax --max_template_date=< max template date>
```

### Selection of benchmark structures

The overall selection process for the cubic structure benchmark is described in Fig. 6. First a list of homomeric tetrahedral, octahedral, and icosahedral assemblies with 12, 24, and 60 chains, respectively, and with a resolution better than 4 Å was compiled from the Protein Data Bank[34]. This list was filtered based on three criteria. First, the subunit in the asymmetric unit was required to have at least one chain without chain breaks. Second, the subunit in the asymmetric unit should not include any non-canonical amino acids (except for selenomethionine, which was treated as a regular methionine). Third, the PDB file should pass through the automatic symmetry detection method (see "Symmetry analysis" section) to create a symmetry definition file used as a template for local assembly experiments. The remaining structures were then clustered at 90% sequence identity with CD-HIT[35] as:

```
cd-hit -i <Input file> -o <output path> -d 0 -c 0.9 -n 5 -G 1 -g 1 -b 20 -l 10 -s .0 -aL .0 -aS .0 -T 4 -M 32000
```

One structure from each cluster with the highest resolution was selected. The remaining structures after this filtering were then sorted based on their sequence length. To save computational time some icosahedral and tetrahedral structures were omitted if they had more than 522 or 544 residues in a single subunit respectively. This resulted in 111 proteins (I:50, O:31, T:30, Fig. 6), whose sequences were predicted with AF and AFM. For AFM, multiple AFM runs were launched corresponding to their symmetric fold interfaces (2-, 3- and 5-fold for I, for example). The total number of AF/AFM predictions was 414, con-

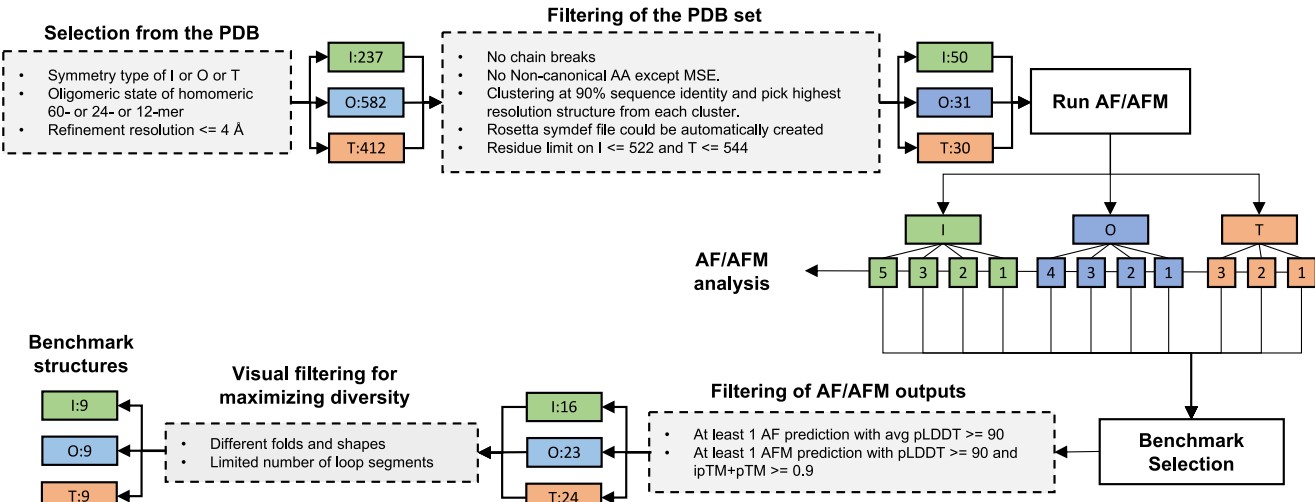

**Fig. 6 | Selection of benchmark and structures for AF/AFM statistics.** Information on each step is elaborated upon in the main text. After each filtering step the remaining PDBs for each cubic symmetry type (green: icosahedral (I), blue: octahedral (O), red: tetrahedral (T)) are shown. The arrows leading away from the Run AF/AFM box indicate which monomeric/oligomeric types were run for the given symmetry type.

tributing 5 models each for a total of 2070 predicted models. AF/AFM was run as described in the previous section. The generated AF/AFM predictions were used in the analysis of the fraction that pass the quality threshold for EvoDOCK assembly as described in the section: The general applicability of the method.

To arrive at the benchmark set, the protein systems were required to contain at least one AF prediction with an average pLDDT ≥ 90 and an AFM prediction (of any oligomer type) with an average pLDDT ≥ 90 and ipTM + pTM ≥ 0.9. Finally, manual inspection of the remaining structures was used to achieve as structurally diverse a set as possible, considering fold, shape, and loop conformations balances within each symmetry type. The final set contained 9 structures from each symmetry adding up to a total of 27 cubic structures.

## Symmetry analysis

A script was developed to automatically analyze the symmetry of native structures with cubic symmetry. The script takes the structure of the complete assembly together with the symmetry type and calculates the 6 parameters describing the degrees of freedom used in the EvoDOCK simulation. The output is a symmetry definition file used to model symmetry in the Rosetta symmetry machinery[10]. To analyze the degree of symmetry of a subcomponent within a natural cubic assembly, we use the make_symmdef_file.pl, script provided by Rosetta and described in Dimaio et al.[10]

## A symmetric version of EvoDOCK

EvoDOCK for heterodimeric docking has been described previously[15]. The program was extensively modified to accommodate symmetry. This included developing (1) an additional set of degrees of freedom, as presented in Fig. 2b. (2) An additional contact-based representation and energy function (CloudContactScore) for identifying clash-free and well-packed subunit orientations. (3) Parameter constraints to enable the use of subsystems during rigid body sampling. (4) A set of rigid body sliding moves to establish contact between subunits during the assembly process. (5) A local search strategy adapted to cubic symmetry to optimize all-atom energy in the system. These developments are described below.

## CloudContactScore (CCS) point cloud representation

To utilize the CloudContactScore score function an atomistic protein structure is turned into a cloud of points. Due to the symmetry machinery, only one subunit must be converted into this representation as symmetry expansion automatically creates all other copies in the assembly. The first step of the point cloud generation is to remove surface residue information on the surfaces beyond β-Carbon atoms, which is done in two steps. The Solvent Accessible Surface Area (SASA) is calculated across all residues and all residues with more than 20 Å SASA-value are labeled as surface residues. Then the SelectResiduesByLayer class in Rosetta, which identifies residue burial based on the number of sidechain neighbors within a cone along a vector from the α-Carbon (CA) and β-Carbon (CB), is used to determine surface residues. All identified surface residues are then changed to alanine residues except for glycines. A final SASA calculation is carried out on this structural representation and all atoms with 0 Å SASA are removed. Only backbone surface atoms of N, C, O, CA, and CB remains at this stage, whose coordinates are used as points in the point cloud representation.

## CloudContactScore (CCS) energy evaluation

There are four main terms in the CCS score function. First, a n_clashes term that penalizes according to the number of clashes ("Clashes" in Fig. 2f). Two atoms are recorded as clashing if the Lennard Jones sphere, as defined per atom in Rosetta, overlaps by more than 20%. Nitrogen−Oxygen interactions can interact through a hydrogen bond and the clash distance is therefore reduced to 1.2 Å. β-Carbon (CB)

Lennard Jones sphere-values are reduced to 1.5 Å to allow for closer interactions on the surface of the structure. To severely penalize clashes, each clash adds a large penalty to the score. Second, a backbone-backbone hydrogen bonding score that uses the hbond_sr_bb and hbond_lr_bb score terms in REF2015[17] is used to model hydrogen bonding ("H-bonds" in Fig. 2f). They model short and large-range hydrogen bond terms, respectively. Lastly, a n_cb_cb_interactions term is used to score the packing-interactions between different chains by counting CB-CB contacts ("CB-CB contacts" in Fig. 2f). The threshold distance for considering CB-CB interactions in the energy calculations is set to 12 Å. Each CB interaction bonus is weighted by the relative connection density of each CB to its subunit. The connection density is defined as:

$$\text{Connection density} = \min \begin{cases} \dfrac{\text{Number of internal CB atoms within 12 Å}}{20} \\ 1.0 \end{cases} \quad (1)$$

Where min means pick the minimum value of the two. The final CCS score is given as a linear combination of the four scoring terms:

$$\text{CCS score} = \text{n\_clashes} + \text{hbond\_sr\_bb} + \text{hbond\_lr\_bb} + \text{n\_cb\_cb\_interactions} \quad (2)$$

## Parametric constraints

The evasion of sampling of nonsensical rigid body orientations is achieved by bounding the rigid body parameters to ranges that maintain the integrity of the subsystem used to model the complete symmetry. For the parameters controlling the radius of the container (z) and radius of the largest n-fold-symmetric system (x), they must have values above 0. A large penalty is added to the score if its parameters are sampled outside this bound using the SQUARE_WELL penalty class in Rosetta with a depth of $10^9$. High values of the λ rotation parameter can also produce nonsensical models. 4 types of symmetry input files are based on the symmetric folds of the cubic structures: 2-, 3-, 4-and 5-fold. The maximum bounds of the λ parameter are set to:

$$\lambda \text{ maximum bounds} : \frac{360}{n} \quad (3)$$

Where n is the n-fold symmetry file used (see Table S1 for which fold symmetry was used for each structure in the benchmark). The bounds are centered around 0 and modeling icosahedral symmetry with a 2-fold symmetry input file, for instance, would yield bounds of [−90, 90] degrees. For the local assembly experiments, half the values of the maximum bounds are used and for global assembly docking the full bounds are used. The other parameters have bounds of ψ: [−40, 40] degrees, Θ: [−40, 40] degrees, φ: [−40, 40] degrees, and x: [−5, 5] Å. These parameters are centered around the template symmetry in the local assembly docking or the values found in the AFM predictions for global assembly docking. For local assembly docking z has the bounds: [−5, 5] Å and for global assembly docking: [0, 1000] Å.

## Sliding moves

All subunits of the cubic system are sequentially slid along the symmetric folds from the highest to the lowest. Tetrahedral structures are slid along their 2- and 3-fold symmetry axis. Octahedral structures are slid along their 2-, 3- and 4-fold symmetry axis. Icosahedral structures are slid along their 2-, 3- and 5-fold symmetry axis. Each fold-symmetric partner is kept fixed relative to each other at each step. The sliding happens in steps of 0.3 Å and ends when clashes are detected according to the CCS n_clashes term or 100 sliding moves have been tried without any clashes emerging. If the structure goes out of bounds it is reverted to the starting configuration.

## Local assembly

The local assembly consists of two main components. The first part is a packing-minimization step that consists of a Rosetta-based sidechain optimization (packing) step followed by a Rosetta-based gradient minimization step that occurs if the energy was decreased by 15 REU (Rosetta Energy Units) during the packing step. Both methods use the Rosetta REF2015[17] score and a Metropolis Criterion to accept the final structure. The second part is a quick rigid body search subroutine consisting of 10 rigid body moves that use the CCS score and Metropolis Criterion to accept. Overall, the packing-minimization step occurs first, followed by the rigid body search, followed again by a final packing-minimization step.

## Ensemble generation

For the local assembly experiments, AFM was run to produce a total of 100 single chain subunits stemming from each cubic symmetry type's respective symmetric folds. AF was run to produce 100 chains. Thus, AFM and AF generated a total of 200 totaling subunits for each system (Fig. S11a). For the global assembly experiments, AFM was run to produce a total of 200 chains stemming only from a single oligomeric subcomponent (Fig. S11b).

AF was run with the flags as described previously, multiple times in succession, to achieve the number of target predictions. AFM was run with the flags as described previously but including: --num_multimer_predictions_per_model to achieve the requested number of target predictions.

The methods internal script used to generate the final ensemble from AF/AFM-predicted subunits (encoded in the script af_to_evodock.py) is shown in Fig. S11c. All AF monomer predictions were first filtered based on their pLDDT (≥90 for the benchmark structures) and all AFM multimer predictions by their pLDDT and ipTM+pTM scores (≥90 and ≥0.9, respectively, for the benchmark structures) to produce an initial ensemble (as in Fig. 3a). The termini were then removed as described in Fig. 3b from the structures in the initial ensemble in a process described here in further detail, using the average values calculated from the structures in the initial ensemble of pLDDT (Avg. pLDDT), secondary structure propensity (SS %) and residue connectivity (connectivity %). The average residue pLDDT is calculated from the AF/AFM output. The secondary structure propensity is calculated with DSSP[36] while the residue connectivity was calculated with a custom function in the af_to_evodock.py script. For this, a contact map is created, and a residue is designated 'disconnected' if it is not in contact (>8 Å) with another residue 10 residue neighbors downstream or upstream to the rest of the structure. Thresholds are set for the three metrics, and the first residue instance that goes above any of the thresholds is recorded. All residues preceding the first instance of the last threshold to be crossed are removed. This process is done going from the N- to C-termini and from the C- to N-termini. The thresholds are set to 90 for the Avg. pLDDT, 70% for connectivity %, and 70% for SS %.

The final ensemble is further reduced to remove structural redundancy by iteratively removing very similar structures. All pairwise RMSD values are calculated and structures with values below 0.1 Å are removed keeping only the model with the best AF/AFM prediction scores. However, if the total size of the final ensemble is less than 50 predictions, the threshold for similarity is reduced by lowering it 0.005 Å for up to 18 steps.

In the global assembly experiments, we extract starting values for some rigid body parameters (ψ, Θ, φ, and x) from the AFM oligomer predictions, and sample around those.

## EvoDOCK simulations

EvoDOCK was run 50 independent times for local recapitulation experiments and 100 independent times for local/global assembly. All runs had a population size of 100 individuals and were run for 50 generations. As we wanted to explore the energy landscape further for some models, the local recapitulation runs of 7Q03 were run 100 times with 100 generations and 6H05 50 times with 100 generations. All simulations ran to completion except for one run for the global assembly docking of 1JH5 and 5A8D. The mutation rate was set to 0.1 and recombination rate to 0.7. For the local and global assembly experiments the rigid body parameters were initially uniformly sampled within their bounds as described in the parametric constraints section. For the global assembly experiments the initial z parameter was however determined by sliding the subunits away and then onto each other again using the CCS score function to stop the sliding when clashes were detected. The template symmetry for the local assembly experiment is derived from the target native structure, while for the global assembly experiments an ideal symmetry is used. Differently from the previous scoring implementation of heterodimeric EvoDOCK is the use of the interface energy (Iscore). We noticed that interactions within a subunit could bias the selection process without improving the overall assembly energy and therefore the Iscore is used as the selection criteria.

## Symmetric energy refinement

One thousand of the best structures based on the interface score (Iscore) were selected from the EvoDOCK runs and k-means clustering from the scikit-learn python package[37] was used to put models into 100 clusters based on their final 6 rigid body parameters. The best models according to the Iscore values within each cluster were selected, to produce a final set of 100 models as inputs to the Rosetta FastRelax method[23].

## Clustering of models

The 100 energy-refined structures were put into 5 clusters based on their 6 rigid body parameters using k-means clustering from the scikit-learn python package[37]. One model from each cluster was selected based on their Iscore resulting in 5 total models. The 5 models are the ones used to evaluate the TM-score[22], Pairwise DockQ score[24], and RMSD.

## TM-score

The TM-score[22,38] was calculated as follows:

$$\text{MMalign <input file> <native file>} - \text{ter } 0$$

With both the input_file and native_file being the full biological assembly.

## Pairwise DockQ score

The Pairwise DockQ score[24] was calculated by summing up the DockQ score for each unique interface respective of each cubic symmetry type: 2-, 3-fold for T; 2-, 3-, 4-fold for O; 2-, 3-, 5-fold for I. For each fold, two chains that form part of the unique interface that matches best according to the RMSD to the two chains of the experimental structure were used in the DockQ score calculation. The two chains were first aligned by their residue numbers as follows:

```
./DockQ/scripts/fix_numbering.pl <predicted dimeric interface>
<native dimeric interface>
```

The output of fix_numbering.pl was then used to calculate the DockQ score as:

```
python DockQ.py <aligned predicted dimeric interface>
<native dimeric interface>
```

The pairwise DockQ score was then calculated by summing up the individual dimeric DockQ scores and normalizing them by their ΔSASA

(Change in SASA when moving the chains away and back into their original position) as follows:

$$\sum_i (DockQscore)_i \times \frac{\Delta SASA_i}{Total\ \Delta SASA} \qquad (4)$$

### RMSD

RMSD was calculated using the Rosetta software[18,28]. The RMSD was calculated on the subsystem as described in the main text compared to the experimental structure. The number of different chain combinations to compare on each evaluation of the full structure is computationally intractable and the subsystem is therefore used. To make sure the full symmetrical system is captured in the RMSD calculation, each symmetrically equivalent configuration of the subsystem is used to calculate the RMSD. This is achieved by rotating around one of the symmetric n-folds n times. For instance, for an icosahedral structure, 5 configurations are tried by rotating the structure around the 5-fold separated by 72 degrees. So RMSD is calculated at 0, 72, 144, 216, and 288 degrees. The lowest value of the RMSD is selected as the report RMSD.

### AFM training set homology calculation

The PDB was culled for sequences with structures with a release date before 2018-04-30, containing between 2–9 chains with no more than 1536 residues. The EMBOSS pairwise sequence alignment software needle[39] was used to calculate all pairwise sequence identities between the culled set and the 111 cubic sequences (including the benchmark set) with the following options:

needle $-$ asequence <seq A> $-$ bsequence <seq B> $-$ sprotein1
$-$ sprotein2 $-$ gapopen 10.0 $-$ gapextend 0.5

where seq A is a sequence of the 111 cubic set and seq B a sequence of the culled PDB set.

### Time complexity

The expected time it takes to reach a given probability of success in the stochastic sampling with EvoDOCK was estimated and the results are presented in Table S2. The strategy is based on the procedure presented in Varela et al.[15] The complete data with RMSD and energy values for each run were resampled. A loop over the number of runs (num_runs: 1–100) was constructed. For each num_runs a random selection of num_runs runs was sampled from the set of 100 runs. This procedure was repeated 100 times for each num_runs. A sample was considered successful if the lowest energy model has a RMSD of less than 4 Å to the native structure at the last generation. To estimate the probability of success, the fraction of sampled data sets among the 100 resampled sets for each combination of num_runs that result in a successful run were calculated. Table S2 shows the numbers of runs and times it takes to reach 80%, 90% and 99% percent success rate. The local assembly benchmark was run on a combination of Intel Xeon E5-2650 v.3 (21 PDBs) and AMD 7413 processors (6 PDBs). The global assembly benchmark was exclusively run on AMD 7413 processors. The Intel CPUs have an approximately 2 times slowdown compared to the AMD CPUs on our benchmark set. Each run of relax takes 10 min in the best-case scenario and up to 4–5 h in the most extreme case on AMD 7413 processors. We estimate that running 25 relax runs should be sufficient in most cases (5 runs per cluster) adding approximately 50 h to the total computational time.

### Reporting summary

Further information on research design is available in the Nature Portfolio Reporting Summary linked to this article.

## Data availability

Source data are provided with this paper. All predicted structures are deposited in Zenodo. The accession codes for all benchmark PDBs are given here: 2ZY2, 2QQY, 3LEO, 7Q03, 6M8V, 6HSB, 4DCL, 2CC9, 4CY9, 3WIS, 5EKW, 5H46, 3N1I, 6H05, 7O63, 7OHF, 3BXV, 7PF9, 1HQK, 1T0T, 4RFT, 4V4M, 1JH5, 6ZLO, 1X36, 2WQT, 7B3Y. Source data are provided with this paper.

## Code availability

The code for running all the simulations in this study is available for download at https://github.com/Andre-lab/evodock[40].

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

## Acknowledgements
This work was supported by the European Research Council (ERC) under the European Union's Horizon 2020 research and innovation program grant number 771820 received by I.A.

## Author contributions
I.A. and J.M conceived the project. J.M. developed the Symmetric Evo-DOCK software, ran all simulations and performed analysis. I.A. contributed to interpreting the results and guiding follow-up experiments. J.M. and I.A. wrote the paper.

## Funding

## Competing interests
The authors declare no competing interests.
