## [Peer Review File · Nature Communications]

Reviewers' Comments:

Reviewer #1:

Remarks to the Author:

The paper presents a method for modeling structures with cubic symmetry based on subunit structures from AlphaFold (AF) and AlphaFold-Multimer (AFM). There is an assumption that the symmetry type is known. The method's main novelty is restricting the search space to fully symmetric structures with predefined symmetry. It relies on Rosetta's local search sampling engine for interface optimization, using a set of specialized symmetry-preserving moves (such as Sliding). Additional optimization of collision detection is achieved using a point cloud surface representation of the subunits. The method is tested for refinement and complete global search scenarios for a set of 21 complexes. It is estimated based on a larger set of complexes that the method can be successful in 40-50% of the cases. The manuscript is very well written and the results are clearly presented.

Comments:

Benchmark construction:

- The benchmark size is very small and likely includes structures that AlphaFold was trained on. As a result, the performance estimate is most likely optimistic. If such a small benchmark is used, at least preference should be given to structures that AF/AFM were not trained on and there are no close homologs in the training set.
- A large fraction (~90%) of PDB structures with cubic symmetry is filtered or clustered. Maybe it is possible to avoid some of the filtering if sequence records are used along with AlphaFold2 to model the structures.
- What are "L segments" in figure 6 caption?
- Is it possible to cluster with a lower sequence identity threshold and avoid visual filtering?

What are the run times of the method? Does it require special resources?

The terminology used to describe the three types of experiments is confusing. Reassembly is a refinement or local search, while complete assembly is ab initio docking or global search. I recommend clarifying this in the text. Also, distinguish better experiments with bound and AlphaFold structures

A table summarizing the 3 types of experiments and results would help to navigate the manuscript

There is a step of trimming disordered termini, what about long loops? can they be trimmed as well?

Is it possible to include examples where the method fails? due to AF/AFM or due to sampling limitations? or maybe additional reasons?

Is it possible to predict the confidence of the predicted assembly?

Reviewer #2:

Remarks to the Author:

This manuscript reports a method of combining AlphaFold / AlphaFold-Multimer with symmetric docking to predict the structures of large symmetric protein assemblies. The method addresses an important problem that has not been solved. The rationale and design of the method is reasonable. It shows some good performance on a benchmark curated by the authors in different situations: the reassembly of using native structures of subunits and the symmetry of the native assembly structure as template and the complete assembly of using only sequence information as

input. However, there is some concern regarding the evaluation and generalization capability of the method that needs to be addressed.

(1) When the authors created the benchmark dataset, they did not consider if the proteins in the benchmark overlap or have substantial similarity with the protein complexes used to train AlphaFold-Multimer, which can lead to the over-estimation of the performance of the complete assembly prediction. The authors should also evaluate the performance on protein assemblies whose sub-components have < 30% maximum sequence identity with any protein complex in the training dataset of AlphaFold-Multimer. The protein assemblies should have little redundancy with the training dataset of AlphaFold-Multimer and they or their components should never occur in the training dataset of AlphaFold-Multimer.

(2) The authors only evaluate their method on self-selected proteins (i.e., homomeric tetrahedral, octahedral, and icosahedral assemblies with 12, 24, and 60 chains), which is not sufficient to accurately estimate the performance of the method on large homomeric assemblies. They should also evaluate their method on the large homo-multimers of the CASP15 experiment to see if their method can generalize well to the kind of homo assemblies that users may frequently encounter.

(3) In the complete assembly prediction experiment, the authors assume knowledge of the correct orientation is known to predict assembly structures. This is not sufficient. The authors should also evaluate their method without assuming any knowledge about the true structures of the assemblies is known to report its performance in a fully blind prediction setting.

(4) Please report the empirical time complexity of the method so that users can know how long it takes to make predictions.

Reviewer #3:

Remarks to the Author:

Summary

In this manuscript, the authors have combined AlphaFold with EvoDOCK for symmetric docking of tetrahedral, octahedral, and icosahedral assemblies. The central theme of the work is to build over AlphaFold structure prediction tool and improve predictions on symmetric assemblies. On a curated benchmark of 21 targets (7 per cubic systems), the authors demonstrate the performance of their strategy and report high-quality structure predictions with best cluster models obtaining a ~50% success rate. The study reports two approaches with their strategy: (a) Reassembly with information from either native template or sub-component information from AlphaFold predicted template; (b) Complete assembly with no prior information (although the authors utilize AlphaFold-multimer structures to extract 6 rigid body).

The present work is interesting because it incorporates AlphaFold as a template generator and builds a symmetric assembly toolkit over it. This work improves on the prediction accuracy for symmetric protein assemblies.

Limitations and Strengths:

The manuscript is succinct. The use of this method is likely to be useful for determination of higher-order symmetric assemblies. However, some sections could improve on clarity to make it easier for the readers. I hereby suggest some technical and scientific comments for the authors to address:

Major Comments:

1. The authors have generated an ensemble utilizing AlphaFold. However, the filtering criteria for selection of targets is stringent such that only highly accurate structures are predicted (pLDDT>90). Does the ensemble generation procedure contribute to diversity or are the structures similar? If the structures are diverse, a note or figure illustrating the diversity would be helpful. For e.g.: comparisons of RMSDs for the ensemble generated for a few benchmark targets can demonstrate that the predictions have diversity.

2. The authors state that the benchmark is limited to structures that have accurate monomeric

structures and some symmetric oligomeric structures. What would happen for cases where the monomeric structures are accurate, but oligomers are incorrect? This would be the ideal test case for complete assembly with no a priori information.

3. For reassembly with AF subcomponents, the authors trim the termini if the confidence is lower. However, there might be regions within the protein core with lower residue pLDDT than the thresholds (for e.g. in Figure 3.B). I believe these irregularities can affect the assembly stage performance as well. What do the authors think? It would be nice to include a note in the section discussing this.

4. For Supplementary Figure S7, please include native energies on the docking plots for reference.

5. The authors define complete assembly as prediction from sequence without a priori information on rigid-body orientation. However, in complete assembly section, line 313-314, the authors state that they extract symmetry information and estimate rigid-body information from the predicted oligomer. This is contradictory to the claim made prior. I understand that the compute time to sample the conformational space would be extensive, however the authors should clarify that in detail (maybe rename it as biased complete assembly?)

6. I would recommend the authors to split Figure 4 in two separate figures. Sections A, B, E (1-3) and F (related pymol figures) could be clubbed as one figure, whereas the rest could be another figure. Additionally, the figure font size is illegible. It would be helpful for the readers if this figure is instead split into two figures with bigger font sizes for readability.

7. The authors do not need to incorporate this in the manuscript, but I am curious if there are approaches to perform complete assembly (without extracting any information from AFM predicted oligomers) on monomeric subunits. Maybe using SymDock in Rosetta or Multi-body symmetric docking in HADDOCK. For the benchmark set of 21 proteins how would the performance of EvoDOCK compare against the other two methods. [This is not necessarily within the scope of the manuscript, so the authors do not need to incorporate it in the main text]

Minor Comments:

1. Introduction, Line 57: Check citation for CASP.

2. Introduction, Line 91: A brief one-line definition of EvoDOCK would be useful in grasping the docking protocol at first.

3. I am unclear about the connection density definition. It seems from the equation on Line 555 that Connection density would always be more than or equal to zero. So, when is the CB atom weight 1.0 as the condition described on Line 556 is never satisfied?

All reviewers: Summary of major changes to the manuscript

We thank the reviewers for making very good suggestion on how to improve our study. The major changes to the manuscript are as follows:

- 6 additional systems have been added to the benchmark, with 27 different assemblies. All added systems have less than 30% sequence identity to any oligomers in the PDB before AFM was trained.
- The global assembly benchmark was rerun with an approach where the orientation is optimized and not assumed.
- Figure 1, 3 and 4 are complemented with new display items for clarity, legibility and to present new results.
- Figure 5 has a new panel F.
- The supplementary materials have been amended with new Figure S3, S8, S9, S10 and Table S1 has been updated while Table S2 is introduced.
- Timing information is presented in Table S2.
- We discovered that the grid summarizing the results, TM-score vs DockQ score, had its axis flipped by mistake. We corrected this now. We also categorize the results in terms of success/not success more on the basis of TM-score since this is more established metric for structure prediction of complexes than the composite DockQ score we utilize here. The stringency of success has been increased to 0.9 for TM score, while we consider DockQ score above and including acceptable. The complete picture of the results is presented in the grid plots.

Reviewer #1

Reviewer comment:

The benchmark size is very small and likely includes structures that AlphaFold was trained on. As a result, the performance estimate is most likely optimistic. If such a small benchmark is used, at least preference should be given to structures that AF/AFM were not trained on and there are no close homologs in the training set.

Author response:

All systems studied in this manuscript are above 9 chains, so were not included in the training of AFM. However, what can happen is that there are homologous proteins that are found with a different symmetry group in PDB that could have been used for training. Also, there can be cases where the same protein has two assembly states in PDB (which indeed, and to our surprise, is the case). Unfortunately, the set used for the training of AFM has not been released and cannot be recreated due to the involvement of random selection of bioassemblies in the creation of the training set. Nonetheless, what we can do is calculate the sequence identity of any of our proteins to any multimeric assembly with less than 9 chains in PDB with a release date before 2018-04-30 (The cutoff date for AFM training data). We report these results in Table S1. Based on this analysis we added 6 new cases with less than 30%

sequence identity to any homooligomer between 2-9 chains in the PDB. We have also added 3 cases where AFM fails to predict a good oligomer, but the monomer is well predicted from this low sequence homology category. Thus 15 out of 30 systems studied in this manuscript are now below 30% identity. Because we benchmark several different modeling scenarios, the total number of assembly simulations presented in this study is over 100, and we are limited by computational resources. But we think the number of cases is sufficient to get a good handle on the performance of the method.

The low homology cases are indeed slightly worse, and we present analysis for expected success rate in this category now as well. We thank the reviewer for pointing this out. Nonetheless, in a practical application of the method homology is beneficial and its difficult to predict which level of homology users of this method would find. But the data will give some indication of the efficiency of the method in different regimes.

Reviewer comment:

A large fraction (~90%) of PDB structures with cubic symmetry is filtered or clustered. Maybe it is possible to avoid some of the filtering if sequence records are used along with AlphaFold2 to model the structures.

Author response:

Unfortunately, we do not fully understand what is meant here. The sequences we used for AF/AFM were extracted from the PDB structures.

Reviewer comment:

What are "L segments" in figure 6 caption?

Author response:

This is a loop secondary structure. We have clarified this now in the figure legend.

Reviewer comment:

Is it possible to cluster with a lower sequence identity threshold and avoid visual filtering?

Author response:

That would indeed provide a more automated selection. We opted against that because it resulted in many assemblies with similar structures being included. For example, we would be getting a lot of systems with ferritin-like assembly or many icosahedra that looked like satellite panicum mosaic virus. To have more structural diversity in the benchmark, we opted for a manual component in the selection. However, the extra systems added in this revision were added purely based on sequence identity.

Reviewer comment:

What are the run times of the method? Does it require special resources?

Author response:

This was indeed missing from the manuscript. With the over 100 assembly simulations presented in this study, we needed to run this on a supercomputer cluster. For the benchmark we have substantially oversampled the number of models to get good statistics on efficiency, but also because we did not know the sampling needs when we launched the benchmark. With the simulation data at hand, we can estimate how long a user must run in order to get a given probability of identifying an accurate model. We have evaluated the median runtime for EvoDOCK it takes to have an 80/90/99% probability to reach a model that is ranked best with the desired quality. The median value is 60h for 90% probability to reach the desired threshold. That is 6h distributed on a 10-core machine, which can be run on a single computer. These results are presented in Table S2.

Reviewer comment:

The terminology used to describe the three types of experiments is confusing. Reassembly is a refinement or local search, while complete assembly is ab initio docking or global search. I recommend clarifying this in the text. Also, distinguish better experiments with bound and AlphaFold structures.

Author response:

Good idea. We have changed the terminology to local and global assembly instead. We want to avoid “local search” because that is used in the manuscript in another context to describe rigid body moves, as well as in the original EvoDOCK paper. To clarify the different modeling scenarios and use of bound and AF/AFM models we have added a new table, in figure 1C. Hopefully this should help.

Reviewer comment:

A table summarizing the 3 types of experiments and results would help to navigate the manuscript

Author response:

A table summarizing the 3 types of experiments is added in Figure 1C.

Reviewer comment:

There is a step of trimming disordered termini, what about long loops? Can they be trimmed as well?

Author response:

The same type of algorithm can be used to remove long internal loops. We did not implement this in this study since it was not required to predict the benchmark set. But we believe this could be important in some cases and plan to introduce this feature in revisions of the method. We have commented on this in the manuscript now.

Reviewer comment:

Is it possible to include examples where the method fails? Due to AF/AFM or due to sampling limitations? Or maybe additional reasons?

Author response:

With the choice of a stringent threshold on the estimated quality of the AF/AFM model, we achieve a very high success rate. But we agree that failures can be valuable. We highlight some failures in Figure S3, and identify them as to do with the energy function, conformational sampling and energy refinement. This include failures that are associated with identification of the wrong orientation, which are presented in Figure S3 and S9. In addition, we have added three cases where AF/AFM is good at predicting the monomeric structure, but where AFM is not able to get the right oligomer structure. When the simulation is started with these orientation estimates, it fails. This is due to sampling issues since we are not getting any low RMSD values. This analysis is added to the results at the end of the section “Global assembly prediction of the structure of cubic systems from sequence”, and the plots are shown in Figure S10. All the energy landscapes are found in Figure S4-S6, so a reader can analyze them themselves as well.

Reviewer comment:

Is it possible to predict the confidence of the predicted assembly?

Author response:

Bryant and Elofsson [PMID: 35273146] developed a metric to predict the DockQ score, called pDockQ. This is presented in Fig. 2 of that study. pDockQ is calculated based on pLDDT and the number of interface residues. We considered reporting this metric, but the average error in the prediction of DockQ scores is quite large. Hence, further developments are required to enable a robust predictive metric for multimers. We reached out to experts in this area, who described ongoing research in the area that we hope will address this shortcoming.

Reviewer #2

Reviewer comment:

When the authors created the benchmark dataset, they did not consider if the proteins in the benchmark overlap or have substantial similarity with the protein complexes used to train AlphaFold-Multimer, which can lead to the over-estimation of the performance of the complete assembly prediction. The authors should also evaluate the performance on protein assemblies whose sub-components have < 30% maximum sequence identity with any protein complex in the training dataset of AlphaFold-Multimer. The protein assemblies should have little redundancy with the training dataset of AlphaFold-Multimer and they or their components should never occur in the training dataset of AlphaFold-Multimer.

Author response:

We agree that this was a shortcoming in how the results were presented. All systems studied in this manuscript are above 9 chains, so were not included in the training of AlphaFold-Multimer. However, what can happen is that there are homologous proteins that are found with a different symmetry group in PDB that could have been used for training. Also, there can be cases where the same protein has two assembly states in PDB (which indeed, and to our surprise, was the case). Unfortunately, the set used for the training of AFM has not been released and cannot be recreated due to process of using a random selection of bioassemblies to populate the training set. Nonetheless, what we can do is calculate the sequence identity of any of our proteins to any multimeric assembly with less than 9 chains in PDB with a release date before 2018-04-30 (The cutoff date for AFM training data). This assumes that all proteins in the set were used for training. We report these results in Table S1. Based on this analysis we added 6 new cases with less than 30% sequence identity to any homooligomer with 2-9 chains in PDB. We have also added 3 cases where AFM fails to predict a good oligomer, but the monomer is well predicted from this low sequence homology category. Thus 15 out of 30 systems studied in this manuscript are now below 30% identity.

Based on this updated set we have now compared success rates in the set with < 30% identity and >30% and presented them in the results section. There is indeed a drop in accuracy, and it was good that we discovered this. We thank the reviewer for pointing this out. We have now calculated the expected accuracy based on this low homology regime, which is presented at the end of the “*General applicability of the method*” section. The results show that the method still predicts models with high accuracy in terms of TM-score, but there are a couple of extra cases where the method fails to produce accurate models. In addition, fewer of the sequences can be predicted by AFM to our selected quality threshold, and we have stratified this in Fig. 5F.

We want to highlight four aspects. First, the set of sequences of homomeric cubic systems with experimental structures is limited so our choice of systems to model is substantially smaller than for other categories of proteins. Second, estimating the degree of homology to known oligomers for future blind predictions is difficult, so the estimated success rates should be taken with a grain of salt. Third, we have shown that our symmetric docking approach works well when an accurate AFM prediction is available, which should improve over time,

and with developments presented in CASP15. Fourth, in a real prediction case, homology is of course beneficial and it is rare to not have any homology.

Reviewer comment:

The authors only evaluate their method on self-selected proteins (i.e., homomeric tetrahedral, octahedral, and icosahedral assemblies with 12, 24, and 60 chains), which is not sufficiently to accurately estimate the performance of the method on large homomeric assemblies. They should also evaluate their method on the large homo-multimers of the CASP15 experiment to see if their method can generalize well to the kind of homo assemblies that users may frequently encounter.

Author response:

It would be highly beneficial to evaluate our method on CASP targets. However, CASP15 does not have any cubic systems. In the multimeric category, restricted to homomers, there were only proteins with cyclical symmetry consisting of dimers (11), trimers (3), hexamer(1), decamer(1), and one 15-mer(1). Systems with cyclical symmetry can already be predicted with AFM or variations of that method with high accuracy. In addition, our methodology is based on the approach to predict complex symmetric assemblies using subcomponents from AFM. AFM is much better suited to predict cyclic oligomers than a method starting from a monomer, in particular for intertwined systems. Hence, docking cyclic oligomers from monomers is typically not the best approach, unless the system is too big for AFM. In that case Rosetta SymDock can be used and EvoDock will not provide many additional benefits beyond computational speed.

Beyond, homomers with cyclical symmetry there are also dihedral and helical symmetrical systems. These can, as we note in the discussion, readily be predicted by extensions to EvoDock. However, the manuscript has been written with a focus on cubic systems and most of the algorithmic developments of the paper are focused on challenges that arise specifically for cubic symmetry. Including other symmetry groups would require a full reconstruction and of rewrite of the manuscript as all figures and method descriptions are focused on cubic symmetry and its challenges. The scope of the manuscript is on cubic symmetry, and we are not presenting the method as a general tool for any symmetry type. Cubic symmetry is the most complex symmetry in nature and requires specific methodology, and we believe that the biological importance of virus capsids makes it a sufficiently important area for the development of custom structure prediction methods. Many researchers also develop cage systems and need tools to model these.

Reviewer comment:

In the complete assembly prediction experiment, the authors assume knowledge of the correct orientation is known to predict assembly structures. This is not sufficient. The authors should also evaluate their method without assuming any knowledge about the true structures of the assemblies is known to report its performance in a fully blind prediction setting.

Author response:

We agree with the reviewer and have rerun all systems for the global assembly prediction with the method where the orientation is learned in the EvoDOCK simulation. This is more efficient than running two independent simulations and picking the case with best energy. The success rate is still very high with this approach, although slightly lower than if we assume knowledge of the correct orientation. All display items for the global assembly benchmark have been changed to reflect this approach, although we still present the result of the original benchmark with known orientation in supplementary materials (Fig. S7 and S8, Table S5). For virus capsids we also demonstrate that electrostatic charge can readily be used to identify the right orientation, in case the energy function result in a misprediction (Fig. S9). With these included, the overall success rates is similar to if we don't assume knowledge of the orientation.

Reviewer comment:

Please report the empirical time complexity of the method so that users can know how long it takes to make predictions.

Author response:

This was indeed missing. We have now added a paragraph on this in the results section and added a Table S2. We have considerably oversampled structures for the benchmark, to give users a better sense of the actual runtimes that is required we have developed a method to calculate the expect runtime to have 80/90/99% probability to reach a model with the desired threshold for structural quality. The numbers are presented in supplementary Table S2. We believe this timing information is beneficial for users of the method in designing the number of independent runs to utilize in simulations.

Reviewer #3

Reviewer comment:

The authors have generated an ensemble utilizing AlphaFold. However, the filtering criteria for selection of targets is stringent such that only highly accurate structures are predicted ($pLDDT > 90$). Does the ensemble generation procedure contribute to diversity or are the structures similar? If the structures are diverse, a note or figure illustrating the diversity would be helpful. For e.g.: comparisons of RMSDs for the ensemble generated for a few benchmark targets can demonstrate that the predictions have diversity.

Author response:

Good suggestion. We have added this analysis to Figure 3C, showing RMSD distributions for the ensembles for each structure. We also show the structural ensembles for 6 cases.

Reviewer comment:

The authors state that the benchmark is limited to structures that have accurate monomeric structures and some symmetric oligomeric structures. What would happen for cases where

the monomeric structures are accurate, but oligomers are incorrect? This would be the ideal test case for complete assembly with no a priori information.

Author response:

This is a good experiment that can showcase the importance of having a good starting point from AFM. We have run three cases where the monomers are predicted accurately with AFM but the oligomers are not well predicted. We have presented the results in the Results section and Figure S10. As expected, we cannot recover the right structure in this scenario and with this level of sampling.

Reviewer comment:

For reassembly with AF subcomponents, the authors trim the termini if the confidence is lower. However, there might be regions within the protein core with lower residue pLDDT than the thresholds (for e.g. in Figure 3.B). I believe these irregularities can affect the assembly stage performance as well. What do the authors think? It would be nice to include a note in the section discussing this.

Author response:

Although we did not have problems with incorrectly predicted loops or loops that interfere with assembly in our benchmark, we agree that this can happen. Our method for trimming termini can be applied to internal loops, and this is an extension we will make in future versions of the method. We have now commented on this in the results section.

Reviewer comment:

For Supplementary Figure S7, please include native energies on the docking plots for reference.

Author response:

We have added these values to Figure S4. The native structure energies are not so useful because the structure has not been rigid-body/sidechain optimized in the energy function. So, they tend to be higher than our predicted models.

Reviewer comment:

The authors define complete assembly as prediction from sequence without a priori information on rigid-body orientation. However, in complete assembly section, line 313-314, the authors state that they extract symmetry information and estimate rigid-body information from the predicted oligomer. This is contradictory to the claim made prior. I understand that the compute time to sample the conformational space would be extensive, however the authors should clarify that in detail (maybe rename it as biased complete assembly?)

Author response:

Our argument for describing it this way is that AFM oligomer prediction is based on sequence only, so the only data that goes into the overall prediction protocol is sequence. We think it is important that a reader is aware that the method only requires sequence information to arrive at the assembly model, even if an oligomer structural model is generated as an intermediate with AFM. We hope this is not deceptive.

Reviewer comment:

I would recommend the authors to split Figure 4 in two separate figures. Sections A, B, E (1-3) and F (related pymol figures) could be clubbed as one figure, whereas the rest could be another figure. Additionally, the figure font size is illegible. It would be helpful for the readers if this figure is instead split into two figures with bigger font sizes for readability.

Author response:

Good suggestion. We did reduce the information in Figure 4 to only the most relevant information, and we moved the pymol figures into Figure 5.

Reviewer comment:

The authors do not need to incorporate this in the manuscript, but I am curious if there are approaches to perform complete assembly (without extracting any information from AFM predicted oligomers) on monomeric subunits. Maybe using SymDock in Rosetta or Multi-body symmetric docking in HADDOCK. For the benchmark set of 21 proteins how would the performance of EvoDOCK compare against the other two methods. [This is not necessarily within the scope of the manuscript, so the authors do not need to incorporate it in the main text]

Author response:

Our method is always based on monomeric input or an ensemble of monomers. AFM is used to initialize parameters within the docking simulation within some boundaries. It is possible to randomize these parameters freely instead, foregoing the AFM predictions altogether. We have tried this for 3 cases and the result is presented in Figure S10. As expected this did not work. SymDock cannot handle this type of symmetry out of the box but can be extended to do so. Our experience with heterodimeric docking shows that the sampling approach used in EvoDock is up to 35 times faster than regular Rosetta Docking. This is also our experience from using versions of SymDOCK in the design of icosahedral capsids. HADDOCK, to the best of our understanding, is limited to cyclical symmetry and cannot handle complex symmetries like cubic symmetry.

Reviewer comment:

Introduction, Line 57: Check citation for CASP

Author response:

Thanks, fixed.

Reviewer comment:

Introduction, Line 91: A brief one-line definition of EvoDOCK would be useful in grasping the docking protocol at first.

Author response:

We have done so.

Reviewer comment:

I am unclear about the connection density definition. It seems from the equation on Line 555 that Connection density would always be more than or equal to zero. So, when is the CB atom weight 1.0 as the condition described on Line 556 is never satisfied?

Author response:

Thank you for pointing this out! That was indeed a mistake on our part. We have fixed the equation and method section to reflect what is actually on.

Reviewers' Comments:

Reviewer #1:

Remarks to the Author:

The authors improved the text and the figures, making it easier to understand the different experiments. The authors also added 9 additional cases to the benchmark to a total of 30 cases (although only 27 are presented through the paper). Overall, the division to test cases that were released before and after AlphaFold was trained helps to get an idea of expected performance. However, it seems that the method will have limited applicability due to two main reasons. First, the method heavily relies on the ability of AlphaFold to produce a good multimer starting configuration for one of the interfaces (as shown by three new cases where AlphaFold was able to produce only monomeric structures), making the method applicable to ~20-30% of proteins without homology to known structures. Second, the runtime requirements of the method are relatively high, limiting applications to specific cases. Running the method on a large dataset of proteins without structures and where the symmetry and stoichiometry are unknown, would not be feasible.

Regarding the metrics: the TM-score captures overall topology. DockQ is a combination of RMSDs and contacts. For this type of approach, which aims to reach high-accuracy models, ICS from CASP, which measures contact accuracy can be a good fit.

It is not clear why pTM and ipTM are used as a sum, rather than separate criteria? If the interface is incorrect, pTM can still be high

Reviewer #2:

Remarks to the Author:

The authors addressed my review comments well in the revised manuscript. The work made significant contributions to the prediction of large protein complex structures.

Reviewer #3:

Remarks to the Author:

Thank you for your responses to the comments. I am in agreement with the proposed changes in the manuscript and the responses.

Response to reviewers:

Reviewer comment:

The authors improved the text and the figures, making it easier to understand the different experiments. The authors also added 9 additional cases to the benchmark to a total of 30 cases (although only 27 are presented through the paper). Overall, the division to test cases that were released before and after AlphaFold was trained helps to get an idea of expected performance. However, it seems that the method will have limited applicability due to two main reasons. First, the method heavily relies on the ability of AlphaFold to produce a good multimer starting configuration for one of the interfaces (as shown by three new cases where AlphaFold was able to produce only monomeric structures), making the method applicable to ~20-30% of proteins without homology to known structures.

Author response:

There are reasons to be more optimistic about the applicability of the method:

- Many researchers are interested in generating an accurate atomistic model of assemblies for systems with some homology to proteins with known structure or may want to study energy landscapes and interpret experimental data with our approach. So, evaluating the applicability of the method solely based on sequences without homology is too restrictive in our opinion. There are also no other methods capable of generating these types of assembly structures even for higher homology cases.
- The accuracy of AlphaFold-Multimer is rapidly improving, with changes introduced in version 2.3 yielding significant improvements. The latest round of CASP also demonstrated that many groups could substantially improve upon vanilla AlphaFold-Multimer. We therefore believe that a larger fraction of systems can already be tackled than we report and that this fraction will continue to rise. It should also be noted that we make artificial restrictions on which template structures to include for the benefit of benchmarking that we will not have to use in a real-case scenario. We have pointed out these considerations in the revised manuscript in the discussion section and at the end of the result section and added a reference describing the performance of AlphaFold-Multimer v 2.3.
- For capsid structures, knowledge of what is inside, and outside is straightforward to gain based on electrostatic charge patterns, which improves accuracy.

Reviewer comment:

Second, the runtime requirements of the method are relatively high, limiting applications to specific cases. Running the method on a large dataset of proteins without structures and where the symmetry and stoichiometry are unknown, would not be feasible.

Author response:

Most potential users of this method are interested in predicting the structure of one or a few systems. For such challenges, it is not a big issue that the results take some time to arrive. Especially if there are no alternative methods available to solve the problem. We agree that high-throughput structure prediction is not feasible with our method. However, we would argue that researchers who utilize a method to predict assembly structure typically have some prior information about the assembly state of their system. In the long run we expect our method to be superseded by a deep learning method that can efficiently utilize symmetry.

Reviewer comment:

Regarding the metrics: the TM-score captures overall topology. DockQ is a combination of RMSDs and contacts. For this type of approach, which aims to reach high-accuracy models, ICS from CASP, which measures contact accuracy can be a good fit.

Author response:

We investigated using ICS, but it turned out to be difficult. There is no software to calculate ICS (to the best of our knowledge), and the citation that presents the metric does not provide sufficient detail to fully replicate it (including definitions of how contacts are calculated). Furthermore, we have not seen any publications that have used ICS outside of CASP evaluation. DockQ on the other hand is widely used in papers describing assembly prediction methods.

Reviewer comment:

It is not clear why pTM and ipTM are used as a sum, rather than separate criteria? If the interface is incorrect, pTM can still be high.

Author response:

pTM + ipTM is used in AFM study to rank models and is used in the colabfold version of AFM as well. In reality, it is not the sum of these terms, but rather a weighted sum. Because the standard output of AFM uses pTM+ipTM to refer to this metric, we also did. But since it is a bit deceptive, we have described the equation for the weighted sum in the main text now instead.